# Qualitative assessment of healthy volunteer experience receiving subcutaneous infusions of high-dose benzathine penicillin G (SCIP) provides insights into design of late phase clinical studies

**Stephanie L. Enkel**[1,2]*, **Joseph Kado**[1,2], **Thel K. Hla**[1,2,3], **Sam Salman**[1,2,4], **Julie Bennett**[5], **Anneka Anderson**[6], **Jonathan R. Carapetis**[1,2,7], **Laurens Manning**[1,2,3]

1 Wesfarmers Centre of Vaccines and Infectious Diseases, Telethon Kids Institute, Nedlands, Western Australia, Australia, 2 Medical School, University of Western Australia, Crawley, Western Australia, Australia, 3 Department of Infectious Diseases, Fiona Stanley Hospital, Murdoch, Western Australia, Australia, 4 Clinical Pharmacology and Toxicology Unit, PathWest, Perth, Western Australia, Australia, 5 Department of Public Health, University of Otago, Wellington, New Zealand, 6 Te Kupenga Hauora Maori, University of Auckland, Auckland, New Zealand, 7 Department of Infectious Diseases, Perth Children's Hospital, Nedlands, WA, Australia

* stephanie.enkel@telethonkids.org.au

## Abstract

## Introduction

Secondary prophylaxis to prevent rheumatic heart disease (RHD) progression, in the form of four-weekly intramuscular benzathine benzylpenicillin G (BPG) injections, has remained unchanged since 1955. Qualitative investigations into patient preference have highlighted the need for long-acting penicillins to be delivered less frequently, ideally with reduced pain. We describe the experience of healthy volunteers participating in a phase-I safety, tolerability and pharmacokinetic trial of subcutaneous infusions of high-dose benzathine penicillin G (BPG)–the SCIP study (Australian New Zealand Clinical Trials Registry ACTRN12622000916741).

## Methods

Participants (n = 24) received between 6.9 mL to 20.7 mL (3–9 times the standard dose) of BPG as a single infusion into the abdominal subcutaneous tissues via a spring-driven syringe pump over approximately 20 minutes. Semi-structured interviews at four time points were recorded, transcribed verbatim and thematically analysed. Tolerability and specific descriptors of the experience were explored, alongside thoughts on how the intervention could be improved for future trials in children and young adults receiving monthly BPG intra-muscular injections for RHD.

**Data Availability Statement:** All relevant data are within the paper and its Supporting Information files.

**Funding:** This research was funded by Cure Kids NZ (7012). JK is supported by a Strep A PhD Scholarship and a Scholarship for International Research Fees at The University of Western Australia (UWA). TKH is supported by a Post Graduate Research Scholarship at University of Western Australia, partly funded by the Althestan Saw Bequest Fund. SE is supported by a Research Program Training scholarship at UWA and a Wesfarmers Centre of Vaccines and Infectious Diseases Top Up scholarship. LM and JC are supported by NHMRC Investigator Awards (GNT1197177 and GNT 1173874 respectively). The funders had no role in study design, data collection and analysis, decision to publish, or preparation of the manuscript.

**Competing interests:** The authors have declared that no competing interests exist.

## Results

Participants tolerated the infusion well and were able describe their experiences throughout. Most reported minimal pain, substantiated via quantitative pain scores. Abdominal bruising at the infusion site did not concern participants nor impair normal activities. Insight into how SCIP could be improved for children included the use of topical analgesia, distractions via television or personal devices, a drawn-out infusion time with reduced delivery speed, and alternative infusion sites. Trust in the trial team was high.

## Conclusion

Qualitative research is an important adjunct for early-phase clinical trials, particularly when adherence to the planned intervention is a key driver of success. These results will inform later-phase SCIP trials in people living with RHD and other indications.

## Introduction

Benzathine penicillin G (BPG) has been used since the 1950s for the prevention and treatment of recurrent episodes of acute rheumatic fever (ARF), caused by repeated *Streptococcus pyogenes* (Strep A) infections [1]. BPG is manufactured by Pfizer (NY, USA) and marketed in Australia as Bicillin® L-A [2]. For secondary prophylaxis (SP) of ARF, the current Australian recommendation is for intramuscular (IM) injection into the outer quadrant of the buttock every 21 to 28 days (1.2MIU dose; 2.3ml) for a minimum of five years, with the aim of preventing rheumatic heart disease (RHD) [3]. Globally, RHD affects more than 40 million people with approximately 340,400 deaths annually, predominantly in resource-limited settings [4]. Additionally, over 471,000 cases of ARF are estimated each year, further placing affected individuals at risk of permanent heart damage [5]. Despite being regarded as a nation with a high standard of health, Australia has an inequitable burden of ARF and RHD in Aboriginal and Torres Strait Islander populations, with annual ARF rates in northern Australia approximating 150–380 per 100,000 among those aged 5–14 years [6]. Sixty-one percent of those with ARF will go on to develop RHD within 10 years, and once diagnosed, 27% will develop heart failure within five years [7]. It is estimated that there are more than 5,000 Aboriginal and Torres Strait Islander people are currently living with ARF or RHD, and that by 2031, there will be a further 8,667 cases, resulting in 663 deaths [8, 9]. Such statistics position RHD as a major driver of cardiovascular inequality between Indigenous and non-Indigenous Australians.

Regular IM injections of BPG are the only proven treatment to prevent recurrent ARF and progression to RHD [10–12]. Oral regimens have more complicated dosing schedules, making adherence difficult and have been shown to offer suboptimal protection [13]. Australian guidelines recommend a minimum adherence to 80% of planned injections (11 of 13 4-weekly doses delivered annually) to provide adequate protection against recurrent ARF [3]. In 2019 only 37% of Aboriginal and Torres Strait Islander Australians prescribed SP achieved this target [14]. Drivers for low adherence to SP include health care access, quality of health care, health care literacy [15] and fear of injection and injection pain [16]. Eurocentric health systems and systemic racism are also detailed as key barriers to achieving high uptake of SP [17]. Pain relieving methods including the addition of lignocaine to BPG [18] and/or the use of a Buzzy® device [19] have resulted in an incremental increase in secondary prophylaxis adherence, however rates remain well below optimal [20]. These barriers and investigations into

patient and service-provider preference have emphasised an urgent need for new longer acting penicillins, with the ideal compound being cost effective, subcutaneously administered and offering a longer duration of protection of three months or more [21]. Prior research into these preferences has also demonstrated that subcutaneous delivery of BPG is safe, tolerable and potentially results in less frequent administration [22].

Qualitative research is a useful accompaniment to clinical trials, providing insight into the impact of complex procedures beyond quantitative pain scores, pharmacokinetics, or efficacy of an intervention [23]. There is a growing emphasis on the need to gain insight into the patient experience, inclusion of their voice and perspective through approaches including semi-structured interviews, focus groups or observations [24]. When appropriately employed, these methods can aid investigators in gathering complementary data that contribute to answering the research question, distilling findings and determining feasibility by assessing acceptability to healthcare providers and patients [25]. For these reasons, we undertook a qualitative sub-study alongside a phase-I trial to describe the experience of healthy volunteers receiving subcutaneous infusions of high-dose BPG (SCIP) [26, 27].

## Methods

The clinical aspects of the SCIP trial have been described elsewhere by Kado et al [26, 27]. In summary, the study assessed the safety, tolerability, and pharmacokinetics of high dose of BPG (Bicillin® L-A, produced by Pfizer [2]) in 24 healthy adult volunteers delivered as a single subcutaneous infusion via a spring-driven syringe infusion pump (Springfusor®30, Go Medical Industries Pty Ltd., Subiaco, Australia), a variable flow control device (VersaRate® Plus, EMED Technologies, El Dorado Hills, California, USA) and a 22G SC catheter (BD Saf-T-Intima™, BD Medical, Mississauga, Ontario, Canada). Participants were assigned to receive either 3.6 MIU (6.9mL, n = 4), 7.2 MIU (13.8mL, n = 10) or 10.8 MIU (20.7mL, n = 10) 3, 6 and 9 times the standard dose respectively, with lignocaine co-administered to reduce pain. On average the infusion took approximately 20 minutes for the largest dose. Safety assessments, pain scores and penicillin concentrations (measured from dried blood spots; DBS) were measured for 16 weeks post dose [28].

For the qualitative component, all study participants were interviewed in English by one of two investigators at four timepoints–immediately prior to their infusion, during their infusion, two hours post-infusion and approximately seven days after dosing. All interviews occurred face-to-face. Interviews were between five and thirty minutes in length, with the first three taking place bedside in the clinical suite the trial was being conducted in, and the last in a private consult room. The semi-structured interviews used a standardised interview guide consisting of a series of open-ended questions regarding reasons for taking part in the study, experience of clinical procedures (including dried blood sample [DBS] collection from finger prick samples, intravenous cannula, and subcutaneous catheter insertion), experience of the infusion, tolerability of the procedure, ability to manage pain in the days post-infusion and suggestions for improvement. Participants were also asked to reflect on how this procedure might be tolerated by children, adolescents and young adults living with ARF [29]. Suggestions of appropriate alterations to improve tolerability were explored.

Interviews were audio recorded and transcribed verbatim. Transcripts were thematically analysed according to the methods detailed by Braun and Clarke [30]. Transcripts were read repeatedly by one investigator (SE) who had conducted half of the interviews with participants and highlighted initial codes using NVIVO 12 [31]. These preliminary codes were discussed with another investigator (JK) who had conducted the remaining interviews and further reviewed to form themes. To improve rigour, results were discussed with the trial team and

those present on dosing day to ensure accuracy of data interpretation. Additionally, a content analysis of all transcripts was undertaken to determine the words or phrases participants most used to describe the infusion experience.

The study was approved by Bellberry Human Research Ethics Committee (2020-12-1348) and registered with the Australian New Zealand Clinical Trials Registry (ACTRN12622000916741). All participants provided written, informed consent prior to participating.

## Results

The 24 participants recruited to the SCIP trial had a median age of 26.9 years and 20 (83.3%) were male. Nine (37.5%) were Caucasian, 8 (33.3%) Asian, 3 (12.5%) Latino, 2 (8.3%) African and 2 (8.3%) of mixed ethnicity. While English was known to be a second language for several participants these data were not captured, however English competency was a requirement for participation in the trial. All consented to their interviews being recorded. As staffing capabilities were constrained on one of the dosing days, questions asked during and two two-hours post infusion were combined and asked during a single recording for four participants. No participants missed an opportunity to be interviewed at timepoints two, three or four and data from 92 recordings were included in the final analysis. The themes discussed below are presented in a manner that aims to demonstrate the participant journey through the study experience. Pseudonyms are used in place of names with basic demographic data provided.

### Financial and altruistic reasons for participation

Most participants had no prior experience with clinical trials, with the SCIP trial being their first. On dosing day, no participant deemed the extensive follow-up schedule to be a possible hindrance or inconvenience. Around one-third of participants stated that they chose to participate due to financial renumeration. Furthermore, as the study required only one day of confinement it was perceived as more convenient when compared against other studies that involved confinement of a week or more. Other participants reported being primarily interested in the research with renumeration offering an added incentive. The 'brand' of the sponsor, an independent academic medical research institute focusing on child health, appeared to be perceived highly among the participants who were motivated to take part in the trial to '*do their part.*'

> '. . .*that is the reason why I volunteered. That is the first thought that came to my mind is children. They deserve a good life.*'

Ben, 54-year-old male.

One participant was directed to the trial after being found ineligible to participate in COVID-19 trials by the contract research organisation (CRO) despite a keen interest. The emphasis on medical research and the need for trial participants amid a pandemic may have increased traffic to the website dedicated to recruitment.

### High tolerance of the dried blood spot procedure

The DBS collection was one of the first procedures participants experienced on the day of their confinement and was repeated three more times in the 12-hours post-infusion and 14 times during follow-up. Overall, the DBS procedure was acceptable to participants, with most tolerating collection well and reporting little to no discomfort or pain.

*'It was very nice. I mean it wasn't that painful. It was like a bee sting. Maybe less, way less'.*

Allan, 31-year-old male.

Others reported minor pain only because they did not know what to expect. A countdown was reported as one way of overcoming this, which anecdotally was a method employed by the trial staff member undertaking follow-ups. For the few participants who experienced a painful first blood spot, it was recalled as being due to needing multiple punctures or inappropriate positioning of the lancet reinforcing the need for staff to be appropriately trained prior to conducting this procedure.

## Low pain resulting from the insertion of the subcutaneous abdominal catheter

The subcutaneous infusion of high-dose BPG was given via a catheter inserted into the lower abdomen, lateral to the umbilicus, without topical analgesia. Most participants reported very little to no pain during this procedure, with many stating they expected much worse given its placement in an area most people rarely receive injections in:

*'It was less pain than everything else we had, with the prick and the arm, it was less. Actually, I am quite impressed. I did not feel anything, I don't know how.'*

Ben, 54-year-old male.

*'Sharp'* and *'scratch'* were the most common terms used to describe how the insertion of the catheter felt to the participant. Some also reported that this stinging continued as the lignocaine (local anaesthetic) was inserted but noted that this was described by the trial staff as to be expected.

*'Oh it was just a small thing and then the pinch stingy feeling and I guess now is fine.'*

Jeremy, 32-year-old male.

## Participant experience of the subcutaneous infusion

Participants were informed that the infusion had the potential to be painful but were aware that given the Phase-1 nature of the trial, additional information could not be provided. While most participants had a tolerable infusion experience, some experienced acute discomfort. Investigator hypotheses as to why this occurred in a subset of patients include catheter placement being too shallow or the infusion rate being too rapid. Approximately one quarter of participants stated the experience was more painful than they had expected.

*'I do not expect the actual infusion to be as painful as it was but I think from like when I saw the other participants they did not go through as much pain as I did, I guess, so I don't think the actual doctor expected it to be that tedious.'*

Douglas, 22-year-old male.

The most stand-out description emerging from the data to explain discomfort felt was a *'step-up'* type pain steadily increasing as the infusion continued that many associated with the pump infusing the next millilitre. Attempts were made to determine exactly where participants

felt the pain of their infusion and while generally abdominal and subcutaneous, a myriad of descriptions were provided.

> *'Every time she* [trial investigator] *called out the millilitres* [leaving the infusion device] *I noticed like that it would hurt again but then it would just kind of like stop hurting. . .Like every time she announced the new millilitre then I would kind of feel it kind of spike a bit and then just subside again. And then she would announce the next millilitre and then it would spike again.'*

Marie, 20-year-old female.

> [After being asked where the pain was]: *'Yeah, like one place like just at the injection site, maybe shooting sideways.'* [Reflecting on the experience two hours post-infusion]: *'It definitely felt on top of, like I could feel it on top of like my abdominal muscles and like in between that layer, like subcutaneous fat.'*

Darren, 26-year-old male.

Participants also expressed that while the initial half or three quarters of the infusion were tolerable and did not hurt, it began to get more intense towards the end of the infusion. Investigators have hypothesised that this is a result of a build-up of pressure in the subcutaneous space of the abdomen, however pain scores collected showed similarities across all dosing cohorts—median (range) pain scores were 2.5 (0–6), 3.0 (0–8) and 2.5 (0–6) for cohorts one, two and three, respectively [26].

> *'Well the first ten minutes of the injection I didn't really feel anything but I think after that, the next ten to fifteen minutes it was quite painful. It was like having a stomach-ache.'*

Scott, 18-year-old male.

Specific descriptors of pain are presented with the size of the word indicating its frequency in the transcripts (Fig 1).

Approximately one-third of participants had a completely uneventful infusion experience with no pain or discomfort felt. Several participants described '*something*' nonspecific, with the most common phrase indicating '*movement*' which could indicate the infusion of BPG into the subcutaneous tissue was able to be felt. There were also descriptors as to whether this felt '*hot*' or '*cold*' with several impressions provided.

> *'It is very little. I feel like there is fluid going in or something. This is stronger now. Yeah, I can feel that but it is not major. I wouldn't even call it pain, it is like I feel something cold, a little bit cold.'*

Ben, 54-year-old male.

Participants expressed physical comfort in the position they remained in during the infusion (supine with a pillow under head). As one participant remarked–'*It doesn't feel like anything is even happening.*' Several participants explicitly stated that they appreciated the privacy that was provided to them, especially considering the location of the procedure and the fact that the abdomen was exposed throughout, however demographic commonalities were not able to be identified to explore this further. One participant who recalled being slightly elevated propped himself up and it made no effect on the infusion indicating that patient preference should always be a consideration.

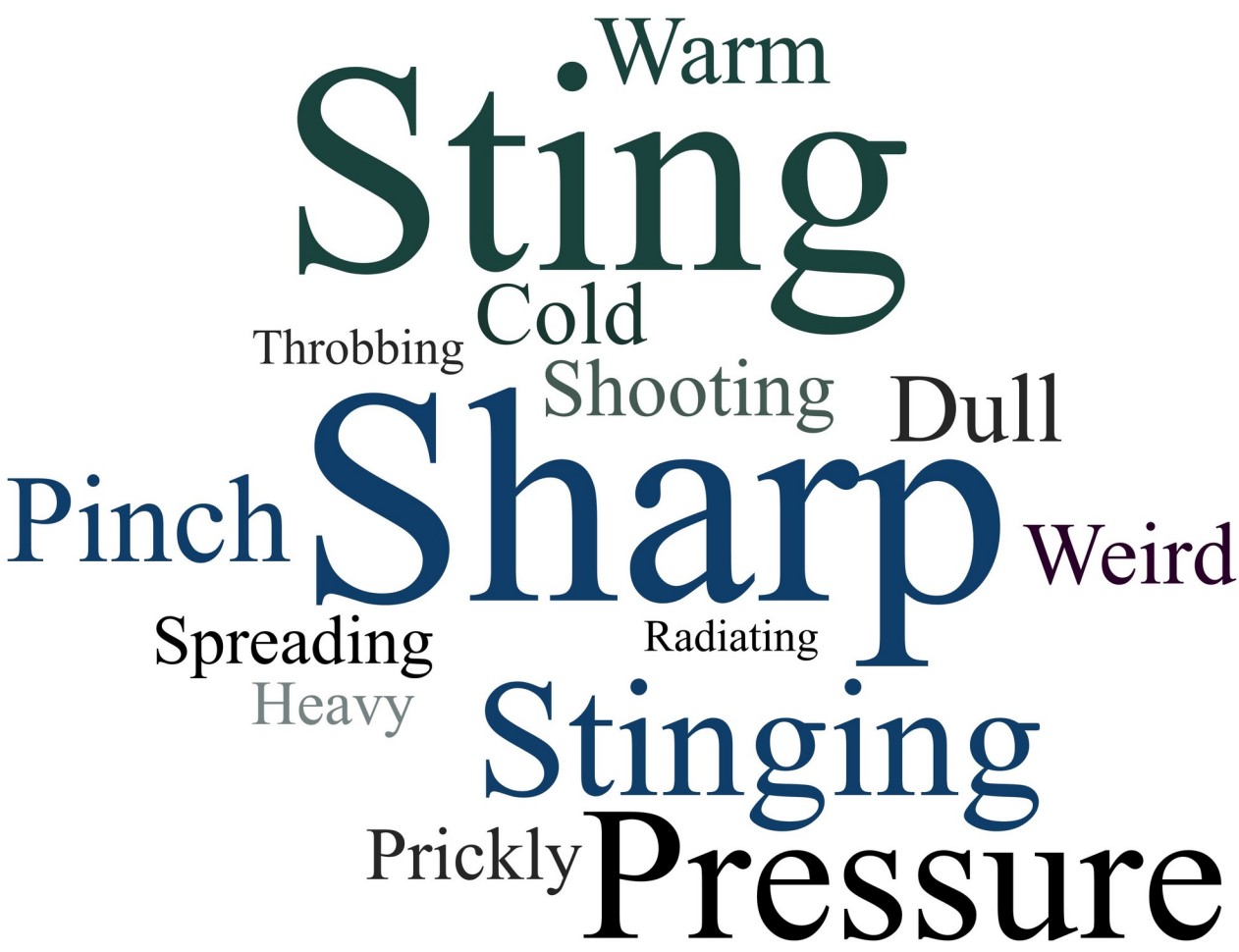

**Fig 1. Word cloud comprised of words used to describe how the infusion was felt by participants.**

Given the infusions occurred in a hospital-ward style room with only curtains for privacy, it was difficult to keep participants unaware about the experiences of others. While participants could not see each other behind the bedside curtains, many stated to hearing the infusion experience of others–inclusive of pain descriptors and verbal reactions—which may have heightened their anxiety immediately prior to the infusion. This sentiment was expressed by those who were beside someone who had an intolerable infusion and is a consideration for further iterations of this trial.

> 'I wasn't feeling much pain, but I guess like, from what I have seen with other participants they seem to be much more in pain compared to me.'

Angus, 21-year-old male.

There was great variability in the acceptability of the time the infusion took. For some–notably those who experienced no discomfort–they felt it could have been shorter. Others thought it could have been prolonged a bit more given their association of pain with infusion speed, while others were accepting of a larger amount of pain over a shorter period. A subset of participants were expecting an injection rather than a prolonged infusion and therefore were probably not mentally prepared for 20 minutes of intense observation with a large trial

team. While the infusion was explained as part of the consent process, this misunderstanding highlights a need to ensure participants wholly comprehend what is being asked of them perhaps via demonstrations or images.

## Experiences in the days and week following the infusion

In the seven days following the infusion, there was no indication that the procedure disrupted the lives of the participants, with most reportedly able to complete day-to-day activities such as getting dressed, driving, and working as normal. Approximately one-third of participants had no concerns. For those who reported pain, it did not appear to be particularly limiting. Many noted that they were expecting a degree of discomfort given they had undergone a medical procedure and adjusted their physical movements accordingly. A subset of participants noted that while they did not feel pain, there was a feeling that '*something*' was present around their infusion site, and they were more aware of their abdominal region than usual.

> *'From personal point of view, I don't think I have had any adverse effects. You know, you get a bit of an injection, you are sore for the next couple of days, it is part of getting the injection.'*

David, 22-year-old male.

> *'Like when I was bending to put my socks on or my shoes, that was like the hardest part because the more I bend the more I felt the pain in my abdominal area. Especially on Sunday* [two days post-infusion] *was the worst day, Monday it started to get better and by Tuesday it wasn't a problem at all.'*

Allan, 30-year-old male.

Skin irritation and bruising were closely examined at each follow-up visit and measured quantitatively by an adapted Skindex-16 questionnaire [32]. Positively the bruising experienced did not overwhelmingly concern participants at Day 7 post-infusion.

> *'I did notice bruising the next day and got to circling. The most extensive bruise occurred up until the second day but afterwards it started to disappear, now you can't even see it anymore.'*

Scott, 18-year-old male.

Of note, at the conclusion of study follow-up two participants were referred for assessment by a dermatologist due to persistent abnormal pigmentation. However, as interviews concluded a week post-infusion it is unclear how this adverse event was experienced.

## Improving the infusion process to ensure acceptability for children

Participants were asked specifically how the infusion process could be better tailored to children, with considerations of their own experience and elements that may have made it more tolerable. By far most participants noted distractions as being critical to ensuring a successful infusion if it was to be undertaken on a child. Suggestions included games, music, blankets, or a support person.

> *'Maybe they need distractions, maybe some TV or some games or something to take their attention away from the injection.'*

John, 43-year-old male.

Despite a local anaesthetic being provided, many participants felt that it was inadequate for a paediatric cohort and additional pain relief would be likely if the procedure was to be undertaken on a child. A topical anaesthetic was also discussed by some participants to reduce the feeling of the needle being inserted into the abdominal region. Furthermore, some participants noted that additional pain relief in the days following the infusion may lead to fewer side effects and pain, particularly as children are expected to be more active. This also included therapies like hot or cold compresses around the area. A few participants stated possibly slowing down the infusion rate as potentially beneficial–especially if paired with a distraction–reflecting on their own experiences of heightened pain when the infusion ran too fast.

*'Well assuming a faster injection would be causing more pain, definitely a slower one would help. It would probably help to have the parents in the vicinity to keep the child company.'*

Andrew, 18-year-old male.

Two participants stated that using two pumps instead of one might be preferable. Both experienced little pain during the initial half of the infusion but discomfort increased substantially towards the end of the infusion. Given neither felt the insertion of the catheter, they perceived this as an avenue worth exploring.

*'Like if you give two injection pumps at the same time, halving the dose because it only hurt going this way, like it didn't hurt this way at all. . . if you can get two needles into a kid somehow. Because to be honest the needle didn't hurt at all.'*

Sarah, 23-year-old female.

One third of participants expressed sentiments that the infusion process in its current form was unacceptable for children and would likely not be tolerated due to the pain experienced. This was particularly relevant given participants were aware the process was to be completed every three months for those who required SP, perceiving pain to be a factor that affected continued treatment. At present, this is a current issue with the standard monthly intramuscular therapy.

## Overall experience with the trial

Despite the pain many experienced, it was promising that most participants reported the overall experience of the SCIP-I trial as positive and held investigators in high regard. Participants expressed feeling able to ask questions as often as required while receiving answers in a timely and easy to understand manner.

*'I think you guys have been really professional. It has been really interesting process and friendly and stuff. That helps a lot to feel like those guys know what they are doing.'*

Allan, 30-year-old male.

Given the numerous trial follow-up appointments and the length of the dosing day, it was reassuring that participants were reportedly able to combine their trial responsibilities with usual commitments and it was positive to note participants being interested and comfortable throughout.

## Discussion

Qualitative research was integrated into the methodology of this trial to better understand the recipient experience of a procedure previously not completed in a volunteer cohort. Given the previously described patient preference for BPG delivery that is less frequent and less painful [21, 33], we weighted equally patient tolerance and pharmacokinetics of the infusion as markers of success. Large doses of BPG delivered by subcutaneous infusions are safe and may be suitable for three-monthly dosing intervals for SP of RHD [26]. This sub-study also found the procedure to be well accepted by participants and mostly tolerable while providing suitable suggestions for improving later study iterations–particularly those in paediatric cohorts.

An element of phase-1 trials which is challenging to address is adequately preparing participants for what to expect, given that by their very nature, there is little prior experience to draw on. This was evident in SCIP specifically for accurate description of pain and the kinds of discomfort likely to be experienced. Concerningly, the qualitative data highlighted two participants were not expecting an extended infusion, apparently interpreting the information provided during screening and the consent process to mean the procedure was a single and quick needle. To address this unexpected gap between 'informed consent' and a clear understanding of the procedures involved, we will be adapting recruitment materials for similar research to include illustrative resources and a greater discussion with study investigators. This combination of verbal, written and even pictorial evidence has been shown to enable higher health literacy and engagement with trial expectations and anticipated outcomes [34] and will be an important aspect to study further with later trial iterations.

SCIP participants were generous in their suggestions on how to improve the study for children, with answers to these questions often the most in-depth answers to any on the guide. These ideas, specifically distractions, a slower infusion time and larger doses of lignocaine anaesthesia, are likely to be integrated into SCIP-II highlighting the importance of understanding the participant experience in early phase clinical trials. A critical question that we were unable to address in this study was how the experience of subcutaneous infusion compares with current practice for SP; i.e., IM injections of BPG every three to four-weeks. The next phase of this research is to undertake a similar study in a cohort of children, adolescents and young adults currently living with ARF (SCIP-II, to be completed in New Zealand) and these questions will be explored with participants through qualitative investigation. In addition, as most pain experienced by participants during the infusion pertained to its speed, alternate infusion methods that allow for more control of injection–including hand pushing of syringes–are under investigation for the second phase and are expected to incorporate greater provision of analgesia.

Significantly more research is required before it can be determined if this nascent procedure is clinically safe and effective and offers desired penicillin concentrations above that required to protect against Strep A infection. However, as investigations progress, we as authors advocate for considerations to be made regarding culturally appropriate techniques for engagement and clear communication with the target population given the burden of ARF and RHD in Australia specifically is experienced by Aboriginal and Torres Strait Islander people. At Telethon Kids Institute–where this research was conducted–the *Guidelines for the Standards for the Conduct of Aboriginal Health Research* were published in 2022, assisting staff to understand what actions and activities they must take throughout their research projects to meet the best practice expectations outlined by both the Institute and the National Health and Medical Research Council (NHMRC) [35]. This eight-step process encourages Aboriginal and Torres Strait Islander involvement well before projects seek ethical approvement for conduct,

and will be used to guide additional study iterations where this population is specifically included.

A strength of this study is that all participants consented to be interviewed and no participant missed a timepoint where qualitative data were collected, likely due to the timepoints of interview (three on the same day) and rapport with the interviewer undertaking questioning at day-7 post interview. Additionally, the same two people conducted all interviews, with the seven-day timepoint completed by the investigator most familiar with participants (having been present at dosing day and the four follow-up appointments since). This allowed for the development of rapport and ultimately, the final interviews were the most in-depth. However, given some adverse events extended beyond 7-days post-infusion, excluding interviews at later follow-up appointments limited opportunities to explore these experiences further. To address this SCIP-II will be involving interviews with participants on infusion day and 28- and 70-days post-infusion. A further study limitation was that all interviews were completed in English which may have repressed the true expression of those for whom this was not their first or most comfortable language. Certain demographic data was not collected (i.e., parental status, preferred language) which impeded a more in-depth analysis of the transcripts and ascertainment of data limitations. Additionally, some transcripts were incomplete where background noise or accents made words unascertainable.

## Conclusion

Qualitative research is a suitable method to be integrated into clinical trials, providing more in-depth information to the tolerability and acceptability of a procedure. The pharmacokinetic data from SCIP support the infusion of high-dose BPG as a possible alternative to monthly IM SP, a finding that would likely be redundant should it prove to be more painful or less tolerable than the current therapy. Our qualitative data demonstrated that the procedure under investigation was acceptable to participants with tolerable discomfort for most. It also identified a subset of participants with more extreme pain, and potential approaches that could be explored to alleviate this. Furthermore, findings from this sub-study have allowed for the development of more detailed participant information sheets for the SCIP-II trial–with suggestions provided by participants of SCIP-I to be incorporated for paediatric and adult cohorts.

## Supporting information

**S1 File. A summary of qualitative themes and supporting quotes.**
(DOCX)

## Author Contributions

**Conceptualization:** Stephanie L. Enkel, Joseph Kado, Thel K. Hla, Julie Bennett, Anneka Anderson, Laurens Manning.

**Data curation:** Stephanie L. Enkel, Joseph Kado, Thel K. Hla.

**Formal analysis:** Stephanie L. Enkel.

**Funding acquisition:** Julie Bennett, Anneka Anderson, Jonathan R. Carapetis, Laurens Manning.

**Investigation:** Stephanie L. Enkel, Joseph Kado, Thel K. Hla.

**Methodology:** Stephanie L. Enkel, Joseph Kado, Thel K. Hla, Sam Salman, Julie Bennett, Laurens Manning.

**Project administration:** Stephanie L. Enkel, Joseph Kado, Thel K. Hla, Laurens Manning.

**Resources:** Stephanie L. Enkel, Joseph Kado, Thel K. Hla, Laurens Manning.

**Software:** Stephanie L. Enkel.

**Supervision:** Joseph Kado, Sam Salman, Julie Bennett, Anneka Anderson, Jonathan R. Carapetis, Laurens Manning.

**Validation:** Stephanie L. Enkel, Joseph Kado, Laurens Manning.

**Visualization:** Stephanie L. Enkel, Laurens Manning.

**Writing – original draft:** Stephanie L. Enkel.

**Writing – review & editing:** Stephanie L. Enkel, Joseph Kado, Thel K. Hla, Sam Salman, Julie Bennett, Anneka Anderson, Jonathan R. Carapetis, Laurens Manning.

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
