## [Decision Letter · Decision Letter 0]

25 Jan 2023

PONE-D-22-31775Qualitative assessment of healthy volunteer experience receiving subcutaneous infusions of high-dose benzathine penicillin G (SCIP) provides insights into design of late phase clinical studiesPLOS ONE

Dear Dr. Enkel,

Thank you for submitting your manuscript to PLOS ONE. After careful consideration, we feel that it has merit but does not fully meet PLOS ONE’s publication criteria as it currently stands. Therefore, we invite you to submit a revised version of the manuscript that addresses the points raised during the review process.

We look forward to receiving your revised manuscript.

Kind regards,

Kovy Arteaga-Livias

Academic Editor

PLOS ONE

Journal Requirements:

  "This research was funded by Cure Kids NZ (7012). JK is supported by a Strep A PhD Scholarship and a Scholarship for International Research Fees at The University of Western Australia (UWA). TKH is supported by a Post Graduate Research Scholarship at University of Western Australia, partly funded by the Althestan Saw Bequest Fund. SE is supported by a Research Program Training scholarship at UWA and a Wesfarmers Centre of Vaccines and Infectious Diseases Top Up scholarship.   LM and JC are supported by NHMRC Investigator Awards (GNT1197177 and GNT 1173874 respectively)." 

Additional Editor Comments:

The reviewers have considered your work to be perfectible, we ask you to take their suggestions into account.

Reviewers' comments:

Reviewer's Responses to Questions

**Comments to the Author**

1. Is the manuscript technically sound, and do the data support the conclusions?

Reviewer #1: Yes

Reviewer #2: Yes

2. Has the statistical analysis been performed appropriately and rigorously? 

Reviewer #1: N/A

Reviewer #2: N/A

3. Have the authors made all data underlying the findings in their manuscript fully available?

Reviewer #1: Yes

Reviewer #2: Yes

4. Is the manuscript presented in an intelligible fashion and written in standard English?

Reviewer #1: Yes

Reviewer #2: Yes

5. Review Comments to the Author

Reviewer #1: To the Authors:

In their manuscript, Qualitative assessment of healthy volunteer experience receiving subcutaneous infusions of high-dose benzathine penicillin G (SCIP) provides insights into design of late phase clinical studies, Dr. Enkle and colleagues describe the results of a qualitative investigation into the tolerability of a subcutaneous infusion of benzathine penicillin in adult volunteers. The protocol was designed to capture meaningful experiential details at multiple time points. The authors write that the results may influence the protocol of a clinical trial, wherein patients (mostly pediatric) will be treated with such infusions in place of monthly IM injections.

The volunteer responses are summarized and tabulated were possible.

Many of the findings revolve around specifics of infusion with respect to speed and anesthesia. It would be interesting to know if the investigators are motivated by these results to define alternative infusion protocols when adverse symptoms are experienced.

Missing from the interviews was a question along the lines of “If you felt it was medically important, would you be willing to attend every three months for a repeat infusion.” One infusion may not indicate long term tolerability. It will be interesting as the underlying clinical trial proceeds to ask patients who have previously been receiving monthly injections how they perceive the subcutaneous infusion as an alternative.

The investigators also noted two subjects who had not understood procedural details despite explanations during the discussion before informed consent. The authors describe modifications that may be made to improve comprehension.

In summary, these results address important details that may inform the process for optimizing patient comfort with and acceptance of a novel process of secondary prophylaxis against RHD.

Specific comment:

Page 11, line 12:

Correct this sentence: “In the week following the SCIP, there was no suggestion the procedure did not appear to disrupt the lives of the participants…”

Reviewer #2: Thank you.

this is a well written manuscript regarding a pilot study of the safety and tolerability of subcutaneous infusions in healthy volunteers.

Specify whether infusions occur by gravity or via a pump.

were interviews only conducted face-to-face or via telephone as well?

Not withstanding that this is a Phase I study, there are references for but limited discussion of potential issues for introducing this to the intended target population. Some further discussion around difficulties and culture-appropriate techniques for engagement and clear communication with a remote indigenous population (children, youth and adults) in view of these results would be helpful.

6. PLOS authors have the option to publish the peer review history of their article (what does this mean?). If published, this will include your full peer review and any attached files.

Reviewer #1: No

Reviewer #2: No

---

## [Author Response · Author response to Decision Letter 0]

14 Mar 2023

Dear Dr. Arteaga-Livias

Thank you for the opportunity to submit a revised version of the original research article entitled “Qualitative assessment of healthy volunteer experience receiving subcutaneous infusions of high-dose benzathine penicillin G (SCIP) provides insights into design of late phase clinical studies” (PONE-D-22-31775).

We are grateful for the time taken by the reviewers to improve this piece of work and as such have made the following changes.

1. (Editor): Please ensure that your manuscript meets PLOS ONE's style requirements, including those for file naming.

The manuscript has been amended to meet PLOS ONE style requirements. 

2. (Editor): Please provide additional details regarding participant consent. 

All participants provided written, informed consent. Additional information to clarify how consent was obtained and what types has been included in the manuscript and online submission statement. 

3. In the ethics statement in the Methods and online submission information, please ensure that you have specified (1) whether consent was informed and (2) what type you obtained (for instance, written or verbal, and if verbal, how it was documented and witnessed). 

All participants provided written, informed consent. This has been included in the Methods of the manuscript and the online submission information. 

4. (Editor): Thank you for stating the financial disclosure: Please state what role the funders took in the study. If the funders had no role, please state: "The funders had no role in study design, data collection and analysis, decision to publish, or preparation of the manuscript." 

These statements have been provided accordingly. 

5. (Editor): In your Data Availability statement, you have not specified where the minimal data set underlying the results described in your manuscript can be found. For more information about our data policy, please see http://journals.plos.org/plosone/s/data-availability. 

A supplementary file with a minimal data set has been provided. 

6. (Reviewer 1): ‘Many of the findings revolve around specifics of infusion with respect to speed and anesthesia. It would be interesting to know if the investigators are motivated by these results to define alternative infusion protocols when adverse symptoms are experienced.’

We thank reviewer 1 for this suggestion and as this is being explored in the second phase of the research (SCIP-II) have included a brief notation as to considerations of strategies. 

7. (Reviewer 1): Missing from the interviews was a question along the lines of “If you felt it was medically important, would you be willing to attend every three months for a repeat infusion.” One infusion may not indicate long term tolerability. It will be interesting as the underlying clinical trial proceeds to ask patients who have previously been receiving monthly injections how they perceive the subcutaneous infusion as an alternative.

As the next iteration of the study is enrolling participants with ARF and RHD who are presently undergoing monthly injections, this question is being asked to serve as a point of comparison. Further information to this point is included in the discussion. 

8. (Reviewer 1): Correct this sentence: “In the week following the SCIP, there was no suggestion the procedure did not appear to disrupt the lives of the participants…”

This sentence has been amended to read as follows: ‘In the seven days following the infusion, there was no indication that the procedure disrupted the lives of the participants.’

9. (Reviewer 2): Specify whether infusions occur by gravity or via a pump. 

Amendments have been made to the methods to clarify that the infusion occurred via a a spring-driven syringe infusion pump (Springfusor®30, Go Medical Industries Pty Ltd., Subiaco, Australia), a variable flow control device (VersaRate® Plus, EMED Technologies, El Dorado Hills, California, USA) and a 22G SC catheter (BD Saf-T-IntimaTM, BD Medical, Mississauga, Ontario, Canada).

10. (Reviewer 2): Were interviews only conducted face-to-face or via telephone as well? 

All interviews occurred face-to-face and this has been clarified in the methodology. 

11. (Reviewer 2): Not withstanding that this is a Phase I study, there are references for but limited discussion of potential issues for introducing this to the intended target population. Some further discussion around difficulties and culture-appropriate techniques for engagement and clear communication with a remote Indigenous population (children, youth and adults) in view of these results would be helpful.

We thank Reviewer 2 for this comment and note it to be an important consideration. A further paragraph has been included in the discussion specific to this point. 

All authors continue to have no conflicts of interest to disclose. The funders had no role in study design, data collection and analysis, decision to publish, or preparation of the manuscript.

 Please address all correspondence concerning to myself at stephanie.enkel@telethonkids.org.au. Thank you for your consideration of this revised manuscript. 

Sincerely,

Miss Stephanie Enkel

On behalf of all authors.

---

## [Decision Letter · Decision Letter 1]

13 Apr 2023

Qualitative assessment of healthy volunteer experience receiving subcutaneous infusions of high-dose benzathine penicillin G (SCIP) provides insights into design of late phase clinical studies

PONE-D-22-31775R1

Dear Dr. Enkel,

We’re pleased to inform you that your manuscript has been judged scientifically suitable for publication and will be formally accepted for publication once it meets all outstanding technical requirements.

Kind regards,

Kovy Arteaga-Livias

Academic Editor

PLOS ONE

Additional Editor Comments (optional):

Reviewers' comments:

Reviewer's Responses to Questions

**Comments to the Author**

1. If the authors have adequately addressed your comments raised in a previous round of review and you feel that this manuscript is now acceptable for publication, you may indicate that here to bypass the “Comments to the Author” section, enter your conflict of interest statement in the “Confidential to Editor” section, and submit your "Accept" recommendation.

Reviewer #1: All comments have been addressed

Reviewer #2: (No Response)

2. Is the manuscript technically sound, and do the data support the conclusions?

Reviewer #1: Yes

Reviewer #2: Yes

3. Has the statistical analysis been performed appropriately and rigorously? 

Reviewer #1: Yes

Reviewer #2: Yes

4. Have the authors made all data underlying the findings in their manuscript fully available?

Reviewer #1: Yes

Reviewer #2: Yes

5. Is the manuscript presented in an intelligible fashion and written in standard English?

Reviewer #1: Yes

Reviewer #2: Yes

6. Review Comments to the Author

Reviewer #1: (No Response)

Reviewer #2: My only minor suggestion would be to include that SC infusion was delivered via spring infusor in the abstract too.

7. PLOS authors have the option to publish the peer review history of their article (what does this mean?). If published, this will include your full peer review and any attached files.

Reviewer #1: **Yes: **Ann F. Bolger MD FAHA FACC

Reviewer #2: No

---

## [Editor Report · Acceptance letter]

20 Apr 2023

PONE-D-22-31775R1 

Qualitative assessment of healthy volunteer experience receiving subcutaneous infusions of high-dose benzathine penicillin G (SCIP) provides insights into design of late phase clinical studies 

Dear Dr. Enkel:

I'm pleased to inform you that your manuscript has been deemed suitable for publication in PLOS ONE. Congratulations! Your manuscript is now with our production department. 

Kind regards, 

on behalf of

Dr. Kovy Arteaga-Livias 

Academic Editor

PLOS ONE